# A Federated Graph Learning Framework With Attention Mechanism and Clustering Algorithm

## Abstract

With the development of the industrial Internet of Things, graph data is also increasing, but these data are held by different clients, and due to client privacy and data security, it is impossible to integrate all the data for unified model training. Federated graph learning can overcome this difficulty very well. It allows clients to participate in the training of the overall model of other clients without revealing their own private data during training, thus protecting the security of clients' private data. However, how to improve the utilization efficiency of client upload parameters to improve the effect of model training and how to process the large amount of initial data owned by clients is an issue that needs to be solved urgently. This paper proposes a federated graph learning framework with attention mechanism and clustering algorithm ($FGL_{AC}$). First, before the client participates in training, a clustering algorithm is used to perform a simple preprocessing operation on the large amount of data held to reduce the overall model training burden and improve training accuracy. Then during the server's process of aggregating model parameters, through the adaptive ability of the attention mechanism, the parameters uploaded by different clients are configured with different weights to obtain the best weight parameters to improve the training effect of the overall model. In order to further verify the effectiveness of $FGL_{AC}$, experimental verification was conducted on different data sets. The results show that in most cases, $FGL_{AC}$ can achieve an improvement of 2.63% - 4.03% compared to other federated graph learning frameworks.

## 1 Introduction

Real world applications of graph data to depict complex relationships between elements of composite objects are widespread. Examples include social networks, citation networks, biochemical networks, and transportation networks. Unlike European data governed by structural principles, graph data have a complex structure and contain a wealth of information. In recent years, graph data research has been a popular topic in the academic community (Liu et al., 2022; Li et al., 2022; Wang et al., 2022). Graph research problems include node classification (Wang et al., 2023b; Zou et al., 2023), graph classification (Chen et al., 2023; Lei et al., 2023), and link prediction (Mi et al., 2023), among others. This paper mainly focuses on the problem of graph classification. Given a set of graphs, the goal of graph classification is to discover the mapping relationship between graphs and class labels and to predict the class labels of unknown graphs. Graph classification is an essential data mining task applicable to a variety of disciplines, for example, molecular graphs are classified in cheminformatics to determine the mutagenicity, toxicity, and anticancer activity of compound molecules (Veličković, 2023; Ji et al., 2023); protein networks are classified in bioinformatics to determine whether a protein is an enzyme and whether it has therapeutic potential for a particular disease(Yin et al., 2023; Zheng et al., 2023). From this perspective, graph classification research is extremely important.

With the rapid development of the Industrial Internet of Things(IIoT), there are more and more graph data in the network environment, and how to process these data has become extremely important. Since the graph neural network can learn the node representation of the non-signature from the graph structure, it has become a hot method for processing graph data in the field of machine learning (Xiao

et al., 2023; Yu et al., 2023; Zhang et al., 2023). The traditional Graph Neural Network(GNN) needs to collect the overall model to aggregate parameters during training (Liu et al., 2023; Yao et al., 2023). However, in IIoT, a large amount of graphic data is owned by different holders, and it is very difficult to integrate all the graphic data. At the same time, due to privacy protection issues in IIoT, a large number of graphic Uniform loading of data for training on GNN is not allowed. This has become a major problem in the application of GNN in IIoT.

As a new distributed machine learning paradigm, federated learning (FL) enables clients to train a globally shared or personalized model in a decentralized manner (Gupta & Gupta, 2023), while not contributing their local data. This property allows FL to be applied to graph data to alleviate data isolation problems and keep each client owning graph data safely. Federated Graph Learning (FGL) is an extension of federated learning on graph neural networks (Fu et al., 2022; Qi et al., 2023; Wang et al., 2023a). It allows the client to train the local model according to the subgraph it owns, and upload the trained parameters to the central server at the same time. The server aggregates the received parameters through the set aggregation rules and then distributes the processed data to the clients participating in the training. After the client receives the parameters, it updates its local model. This process is iterated until the model converges completely.

Most of the traditional federated graph learning paradigms use FedAvg (McMahan et al., 2017) when the server aggregates client parameters. The client participating in the training uses the initialized global model parameters to initialize the local model. Multiple rounds of gradient descent and multiple updates to the parameters ensue; the client then transmits the parameters of the local model to the server, and the central server aggregates the local model parameters into global model parameters through a weighted average aggregation strategy (Fu et al., 2023; Silva et al., 2023; Ghimire et al., 2023). FedAvg can implement distributed GNN model training very well, but he does not consider the weight impact of different client local parameters on all client model training, that is, in the actual federated graph learning process, the data uploaded by each client Parameters have different influences; how to achieve different degrees of learning of parameters of different clients according to different weights when the server performs parameter aggregation. And in IIoT, the number of local data sets owned by the client is very large. If the data is not processed and directly trained on the model, it may be expensive. At the same time, unprocessed data sets may have a certain impact on the training effect, which is something that the traditional federated graph learning paradigm has not considered.

In order to resolve the aforementioned issues, this paper proposes a federated graph learning framework that uses a clustering algorithm to preprocess client local data and uses an attention mechanism to assign different weights to the local parameters of different clients. $FGL_{AC}$ first allows the client to use a clustering algorithm to perform a preprocessing operation on the data set it owns before training the model based on local data, so as to better process downstream tasks. At the same time, in the process of aggregating local parameters of the client by the central server, adaptive attention parameters are added. After the server receives the local aggregation parameters from different clients, it assigns different aggregation weights to different local parameters according to the different training effects of the clients, and obtains a global parameter that is most suitable for all clients participating in the training. This is used to improve the local model training accuracy of the client, thereby improving the training effect of the overall framework. The following are the primary contributions of this paper:

- Before performing local model training, the client uses a clustering algorithm to perform a preprocessing operation on its own data set, and uses the preprocessed information as auxiliary information for downstream task execution. This can decrease the overall training burden and enhance model training's effectiveness.

- In the process of local parameter aggregation uploaded by the central server to the client, the attention mechanism assigns different aggregation weights to the uploaded parameters according to the good or bad training effect of different clients, to get a global parameter that is most suitable for the target client and drive the overall training effect.

- Through a large number of experiments and experimental results, it is shown that $FGL_{AC}$ can have a better training effect than the traditional federated graph learning framework.

## 2 RELATED WORK

### 2.1 GRAPH SELF-ATTENTION MECHANISM

Graph self-attention mechanism: The idea of attention first appeared in the field of computer vision, trying to reduce the computational complexity of image processing while improving performance by introducing a model that only focuses on a specific region of the image rather than the entire image(Xia et al., 2022; Cui et al., 2022; Cheng et al., 2023). Through the continuous improvement of attention technology, it can be popular in various tasks, such as text classification(Ahmed et al., 2022), image description(Ding et al., 2023), sentiment analysis(Peng et al., 2023), speech recognition(Zeyer et al., 2023) and so on. With the continuous development of technology, graph-structured data continues to appear in different neighborhoods. Much useful information can be obtained from graph-structured data by representing data as a graph, which captures entities (i.e., nodes) and the relationships between them (i.e., edges). However, in the real world, graph-structured data may contain a lot of complex information and may also contain a lot of irrelevant noise information, which makes it difficult to effectively process graph-structured data. An effective way to solve this problem is to add "attention" to the research on the number of graph structures(Wu et al., 2023; Ahmad et al., 2023; Fan et al., 2023). The attention mechanism enables the model to concentrate on the task-relevant portions of graph-structured data, thereby improving its decision-making.

### 2.2 CLUSTERING ALGORITHM

Clustering Algorithm: The spectral clustering algorithm is an algorithm for clustering that is founded on the theory of spectral graphs. It is predominantly divided into two classes: iterative spectral clustering algorithms and multi-path spectral clustering algorithms, respectively, based on the SM algorithm (Shi & Malik, 2000) and the NJW algorithm (Ng et al., 2001) to represent. The concept of spectral graph partitioning inspired the concept of a spectral clustering algorithm. In accordance with sample similarity, spectral clustering generates an undirected weighted graph. Consider the sample points to be the graph's vertices, and the weight of the edge between them to be their similarity. The spectral graph division of an undirected weighted graph divides it into multiple subgraphs, which corresponds to the clustering procedure of the clustering algorithm. For spectral graph partitioning, the choice of graph partitioning criteria will have a direct impact on the partitioning outcomes. Typical graph partitioning criteria consist of canonical cut sets, minimum cut sets, average cut sets, and proportional cut sets, among others (Pang et al., 2018). In contrast to spectral graph partitioning, the spectral clustering technique takes the continuous relaxation form of the problem into consideration and transforms the graph segmentation issue into a spectral decomposition issue related to locating similarity matrices (Mei et al., 2023). In the federated graph classification task, the amount of data required for federated learning is very large. If the client does not preprocess the data locally, the amount of communication and calculation for the entire federated learning system will be huge. The spectral clustering algorithm can handle graph-structured data very well (Klus & Djurdjevac Conrad, 2023; El Hajjar et al., 2022), and federated graph learning itself is a distributed learning method, which is also a very suitable application field for the spectral clustering algorithm.

## 3 METHODOLOGY

### 3.1 OVERVIEW

The $FGL_{AC}$ framework proposed in this article uses the spectral clustering algorithm to preprocess the graph mechanism data owned by the client before the client participates in federated training and performs clustering according to the specified clustering range to provide good Input data, improve the efficiency and communication performance of federated graph learning, and reduce communication overhead. Secondly, in server aggregation, through the self-adaptive attention mechanism, different proportion weights are assigned to the parameters uploaded by different clients, and a set of specific weight parameters is saved for each client, improving the client model. At the same time, it can also affect the global model training of federated graph learning. The example diagrams of $FGL_{AC}$ are shown in Fig. 1 and 2, where Fig. 1 is the preprocessing of the data by the spectral clustering algorithm and Fig. 2 is the process of federated learning. The algorithmic process of

$FGL_{AC}$ is shown in Algorithm 1. Table 1 shows the meaning of each variable used in the algorithm 1.

Table 1: Notations & Explanations

| Notations | Explanations |
|---|---|
| $K$ | Number of participating enterprises |
| $\mathbb{C}$ | Collection of training customers, where $\mathbb{C} = \{C_1, C_2, ..., C_k\}$ |
| $\mathbb{D}$ | Training data collection, where $\mathbb{D} = \{D_1, D_2, ..., D_k\}$ |
| $\mathbb{G}$ | The quantity of global iterations |
| $\mathbb{L}$ | The quantity of local iterations |
| $S$ | Server |
| $Z$ | Global parameters |
| $z_k$ | Local parameters of client $C_k$ performing training tasks on dataset $D_k$ |
| $W$ | Similarity matrix |
| $D$ | Degree matrix |
| $L$ | Laplacian matrix |

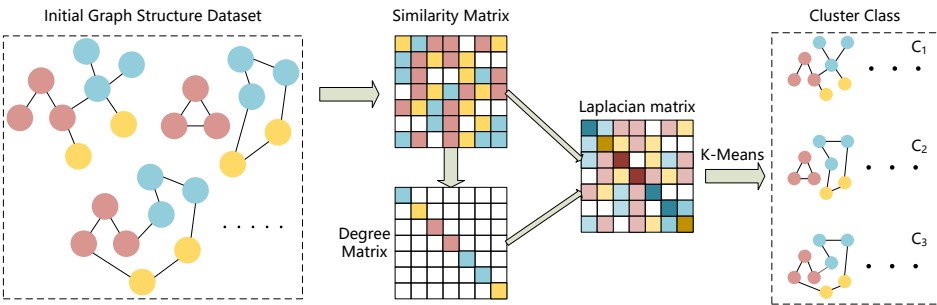

Figure 1: **The initial data set is preprocessed using a spectral clustering algorithm.**

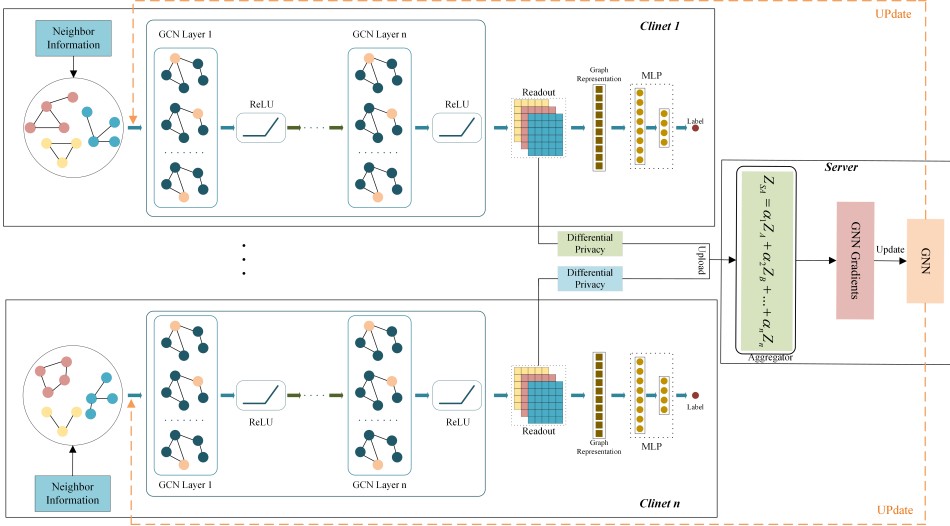

Figure 2: **Federated graph learning framework with attention mechanism and spectral clustering.**

---

**Algorithm 1** A Federated Graph Learning Framework With Attention Mechanism and Spectral Clustering

---

**Input:** $\mathbb{C},\mathbb{D},\mathbb{G},\mathbb{L},S,Z,z,\mathbb{L}(z)$
**Output:** Final Global parameter $Z$
1: Initialize $\mathbb{G},\mathbb{L},Z,z$;
2: # Local dataset preprocessing
3: Calculate the similarity matrix $W$ by (1);
4: Calculate the degree matrix $D$ by (3);
5: Calculate the Laplacian matrix $L$ by (4) using $W$ and $D$;
6: Use (6,7) for clustering operations;
7: **for** $g = 1$ to $\mathbb{G}$ **do**
8:    # Training for local model
9:    **for** each $C_k \in \mathbb{C}$ in parallel **do**
10:      From server $S$ download global parameter $Z$;
11:      **for** $l = 1$ to $\mathbb{L}$ **do**
12:        Classification by (13 or 15) training data set;
13:        Calculate the loss $\mathbb{L}_k(z)$ on $\mathbb{D}_k$;
14:        Update local parameter $z^k_{(g,l+1)}$;
15:      **end for**
16:      Update loss $\mathbb{L}_k(z)$ and parameter $z^k_{(g,\mathbb{L})}$ from $C_k$ to $S$;
17:    **end for**
18:    # Global aggregation
19:    Receive model parameters $z$ from clients participating in training;
20:    Distribute the global parameter $Z_g$;
21: **end for**
22: **return** $Z$;

---

## 3.2 DATA PREPROCESSING

In this article, the spectral clustering algorithm is utilized as a data preprocessing strategy, so in this section, how to use the spectral clustering algorithm to implement the data preprocessing function is described in detail. The idea underlying the spectral clustering algorithm is to take into account every point of data in space, and to connect these points with edges. Edges between points that are far away have low weights, and edges between points that are close have high weights. In this paper, each sub-graph in the data set is regarded as a point in the space, and all the graphs are clustered by the spectral clustering algorithm through a false edge. Given a data set $G = \{g_1, g_2, ..., g_n\}$, the specific operation is as follows.

The initial step is to create the similarity matrix $W$, where the Euclidean distance is used to calculate the distance between two sample points. Utilize the $KNN$ algorithm for traversing every one of the sample points and select the $k$ nearest points as neighbors for each of the samples, only $w_{ij} > 0$ between the $k$ points closest to the sample, and $w_{ij}$ only if the two points are $k$ neighbors to each other. Specifically, it can be expressed as:

$$w_{ij} = w_{ji} = \begin{cases} 0 & g_i \notin KNN(g_j) \ or \ g_j \notin KNN(g_i) \\ \exp(-\frac{||g_i - g_j||^2_2}{2^{\psi^2}}) & g_i \in KNN(g_j) \ and \ g_j \in KNN(g_i) \end{cases} \tag{1}$$

Where, $||g_i - g_j||^2_2$ is the Euclidean distance between two sample points, $\psi$ is the scale parameter, and $W$ changes with the value of $\psi$.

Next is to construct a degree matrix $D$. In this work, for two sample points $g_i$ and $g_j$ with correlation, $w_{ij} > 0$, and for two sample points $g_i$ and $g_j$ without correlation, $w_{ij} = 0$. Therefore, for any sample point $g_i$ in the set, its degree $d_i$ can be defined as the sum of all weights associated with it, namely:

$$d_i = \sum_{j=1}^{n} w_{ij} \tag{2}$$

Using a definition of the degree of each sample point, a degree matrix $D$ can be obtained, which is a matrix of diagonals, and only the primary diagonal has values, corresponding to the degree of the $i^{th}$ point in the $i^{th}$ row, which can be expressed as:

$$D = \begin{pmatrix} d_1 & \cdots & \cdots \\ \cdots & d_2 & \cdots \\ \vdots & \vdots & \ddots \\ \cdots & \cdots & d_n \end{pmatrix} \tag{3}$$

The similarity matrix $W$ and degree matrix $D$ constructed above can be used to construct the Laplacian and standardized Laplacian matrix. Specifically, it can be expressed as:

$$L = D - W \tag{4}$$

$$\tilde{L} = D^{-\frac{1}{2}} L D^{-\frac{1}{2}} = D^{-\frac{1}{2}} (D - W) D^{-\frac{1}{2}} = I - \tilde{W} \tag{5}$$

where $L$ is the Laplacian matrix and $\tilde{L}$ is the normalized Laplacian matrix. Compute the eigenvalues of $\tilde{L}$ and arrange them from smallest to largest, calculate the eigenvectors of the first $k$ eigenvalues, and form the $k$ eigenvectors into a matrix $U = \{u_1, u_2, u_3, ..., u_k\}, U \in R^{n*k}$ to create a new solution space. Use $K - means$ algorithm for clustering on the new solution space, let $x_i \in R^k$ be the vector of $i$-th row , where $i \in (1, 2, ..., n)$, $U = X = \{x_1, x_2, ..., x_n\}$ , then the objective function can be expressed as:

$$d(X, C_i) = \sqrt{\sum_{j=1}^{d} (X_j - C_{ij})^2} \tag{6}$$

Where, $X$ is the data object, $C_i$ is the $i^{th}$ clustering center, $d$ is the data object's dimension, and $X_j$ and $C_i j$ are the $j^{th}$ attribute values of $X$ and $C_j$, respectively. The formula for calculating the sum of squared errors for the entire dataset is:

$$SSE = \sum_{i=1}^{k} \sum_{X \in C_i} |d(X, C_i)|^2 \tag{7}$$

Where, $k$ represents the number of clusters, and the magnitude of $SSE$ shows the clustering result's quality. Then the clustering results are mapped back to the space of the original solution, which is used as the input of the graph classification task.

## 3.3 FEDERAL ATTENTION

In the traditional server aggregation process, most of the aggregation uses FedAvg or FedProx, which does not take into account that the parameters uploaded by some clients have a greater impact on the aggregation of the server, while the parameters uploaded by some clients have less influence. Suppose three clients are participating in federated graph training, each client has a set of graph structure data $G_i = (g_{i1}, g_{i2}, ..., g_{in})$, the client obtains a set of parameters $Z_i$ through local training, and then uploads the parameters to the central server for aggregation. In this paper, the server uses the attention mechanism to aggregate the parameters uploaded by the client and assigns different weights to the parameters uploaded by each client according to the contribution to the server aggregation through the attention mechanism. The objective function can be expressed as:

$$Attention(c_i, c_j) = \frac{\exp(\text{LeakyReLU}(a^T[Wc_i||Wc_j]))}{\sum\limits_{k \in \mathcal{N}_i} \exp(\text{LeakyRuLU}(a^T[Wc_i||Wc_k]))} \tag{8}$$

Where, $c_i$ and $c_j$ represent the feature vectors of the current client and another client, respectively. $W$ is the learnable weight matrix used to map the input features into the attention space. $[Wc_i||Wc_j]$ means to connect the parameters uploaded by the client into a new feature vector, where $||$ means the connection operation of the vector. $a$ is a learnable parameter vector for computing attention

weights. $\mathcal{N}_i$ represents the set of other clients except client $c_i$. A set of different attention weights can be obtained through the formula 8. For each client, this paper uses different weight coefficients to aggregate the parameters uploaded by the current client and other clients. Assuming that for the client set $C = (C_1, C_2, C_3)$, the uploaded parameter sets are $Z_1$, $Z_2$, $Z_3$ respectively, the objective function can be expressed as:

$$Z_C = \alpha Z_1 + \beta Z_2 + \delta Z_3 \tag{9}$$

Where, $\alpha$, $\beta$, and $\delta$ are a set of weight parameters calculated by different clients acting as target clients through formula 8. By assigning different weight parameters, the parameters uploaded by the client that contribute more to the client aggregation can occupy a larger proportion of the aggregation process. The advantage of this is that the trained client may adjust different weight parameters to drive poorly trained clients to achieve better training of the global model. During each iteration of the federation, the client will aggregate different parameters through this method, and then redistribute them to the clients participating in the training, and the clients will conduct a new round of training according to the new parameters received. This process has been iterated until the global model converges.

At the same time, the server will also save the parameters uploaded by each client, so that there are clients in the same group of clients that have not participated in the federation training. At this time, the client can update the parameters of the training of other clients in the same group to the clients that did not participate in the training, so that the clients that did not participate in the training can also be affected by the training parameters of other clients.

## 4 EXPERIMENTS

This section only analyzes the results of 4.1 Graph Classification, 4.2 Ablation Experiments, and 4.3 Comparison between Distributed Training and Centralized Training. The Experiment Setup (such as Datasets, Baselines, Implementation Details) are shown in Appendix A.3, Performance Comparison in A.4, Visualization in A.5, and Attention Parameter Analysis in A.6.

### 4.1 GRAPH CLASSIFICATION

Table 2: **Graph classification results (%). (Index: Accuracy, F1 Bold: best.)**

| Datasets | Type | Index | GCN-FedAvg | SAGE-FedAvg | GCN-FedProx | SAGE-FedProx | $FGL_{AC}$ |
|---|---|---|---|---|---|---|---|
| MUTAG | balance-no-overlap | Accuracy | 86.48 | 81.58 | 84.21 | 84.21 | **86.84** |
| | | F1 | **84.41** | 76.97 | 80.81 | 78.16 | 83.55 |
| | unbalance-no-overlap | Accuracy | 78.95 | 76.32 | 78.95 | 73.68 | **81.58** |
| | | F1 | 70.88 | 62.62 | 72.86 | 50.52 | **75.44** |
| | balance-overlap | Accuracy | 81.58 | 73.68 | 81.58 | 78.95 | **84.21** |
| | | F1 | 76.97 | 42.42 | 75.44 | 70.88 | **80.81** |
| | unbalance-overlap | Accuracy | 84.21 | 84.21 | 84.21 | 84.21 | **86.84** |
| | | F1 | 81.73 | **81.73** | 78.16 | 73.73 | 79.23 |
| ENZYMES | balance-no-overlap | Accuracy | 41.67 | 35.00 | 42.50 | 35.00 | **43.33** |
| | | F1 | 41.83 | 31.88 | 40.70 | 38.85 | **42.15** |
| | unbalance-no-overlap | Accuracy | 36.67 | 38.33 | 44.17 | 38.33 | **44.17** |
| | | F1 | 33.67 | 37.95 | 41.03 | 36.36 | **41.03** |
| | balance-overlap | Accuracy | 40.83 | 37.50 | 42.50 | 40.00 | **45.00** |
| | | F1 | 37.78 | 37.26 | 39.91 | 39.18 | **44.70** |
| | unbalance-overlap | Accuracy | 37.50 | 35.00 | 36.67 | 37.50 | **37.50** |
| | | F1 | 34.62 | 32.99 | 35.65 | **36.73** | 33.50 |
| PROTEINS | balance-no-overlap | Accuracy | 72.65 | 69.64 | 73.21 | 69.64 | **75.00** |
| | | F1 | 70.66 | 68.84 | 73.00 | 67.30 | **73.33** |
| | unbalance-no-overlap | Accuracy | 70.85 | 69.64 | 75.00 | 71.43 | **75.00** |
| | | F1 | 67.86 | 69.63 | 69.52 | 71.39 | **74.97** |
| | balance-overlap | Accuracy | 72.20 | 71.43 | 76.79 | 74.11 | **78.57** |
| | | F1 | 69.79 | 71.42 | 76.60 | 71.65 | **77.87** |
| | unbalance-overlap | Accuracy | 70.40 | 73.21 | 76.79 | 78.57 | **78.57** |
| | | F1 | 67.84 | 72.66 | 76.17 | 78.12 | **78.12** |

The performance evaluation of $FGL_{AC}$ on the federated graph classification task on the above three datasets is shown in Table 2. In this work, the classification tasks of three clients and one central server are simulated. During the experiments, each dataset is divided into four cases for training as described in subsection 4.1.1 to test the classification performance of $FGL_{AC}$. Based on the above results, it can be concluded that:

- The $FGL_{AC}$ framework shows relatively good experimental results on most datasets, which shows the effectiveness of adding a spectral clustering algorithm and attention mech-

anism in the process of federated graph learning. First, the client can use the spectral clustering algorithm to preprocess its local data to relieve the pressure of communication. Secondly, the server uses the attention mechanism when aggregation, and can use the client parameters with better training effect to drive the poorer training effect client.

- Compared with the traditional federated graph learning, the $FGL_{AC}$ proposed in this paper has a good performance improvement in the results, which meets the expected effect. Even in the worst case, all client training effects are the same, and $FGL_{AC}$ will degenerate into FedAvg without affecting the overall training results. However, once the training results of some clients are slightly better, $FGL_{AC}$ can use the better training parameters to optimize the overall training results.

- In $FGL_{AC}$, when the client uses the attention mechanism to aggregate parameters, all attention parameters are learned by themselves, and the parameters used for aggregation are constantly adjusted through each round of iterations. The experimental results and related theories prove that the self-learned attention parameter improves the aggregation effect of the server and does not have a negative impact on the overall training. Even in the worst case, $FGL_{AC}$ uses consistent parameters that convert to FedAvg.

## 4.2 ABLATION EXPERIMENT

In order to further verify the influence of the spectral clustering algorithm and attention mechanism on the overall training in $FGL_{AC}$, ablation experiments are performed on $FGL_{AC}$ in this subsection. Compared with the traditional GCN-based FedAvg federated graph learning algorithm, the small data set MUTAG is used as the test sample, and Accuracy is used as the indicator to compare the two situations of unbalance-no-overlap and balance-overlap. The specific results are depicted in Fig. 3 and Fig. 4 shows.

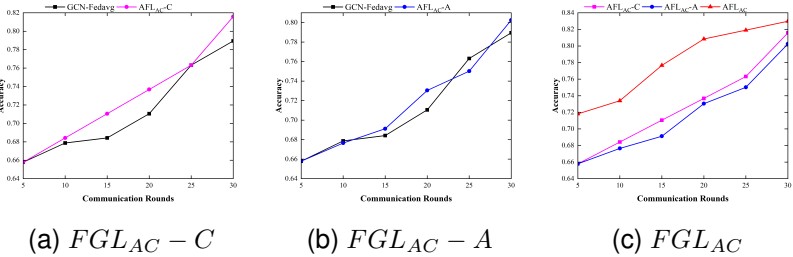

(a) $FGL_{AC} - C$      (b) $FGL_{AC} - A$      (c) $FGL_{AC}$

Figure 3: **Ablation experiment in unbalance-no-overlap environment.**

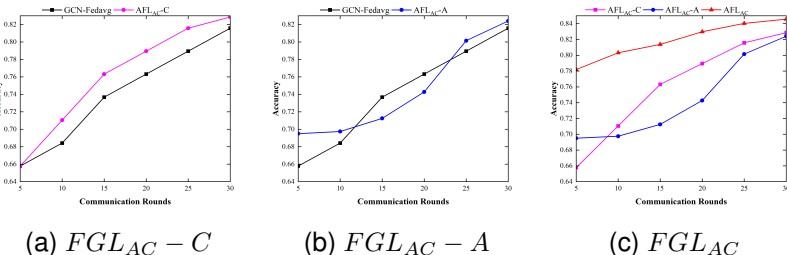

(a) $FGL_{AC} - C$      (b) $FGL_{AC} - A$      (c) $FGL_{AC}$

Figure 4: **Ablation experiment in balance-overlap environment.**

In this paper, the $FGL_{AC}$ framework is split into three categories, namely, removal of client nodes for preprocessing data using spectral clustering algorithms ($FGL_{AC} - C$), removal of servers for parameter clustering using the attention mechanism ($FGL_{AC} - A$), and the complete $FGL_{AC}$ for ablation experiments. Fig. 3 and Fig. 4 are the comparisons between the three $FGL_{AC}$ frameworks

with Accuracy as the index and the traditional GCN-based FedAvg federation algorithm in the case of unbalance-no-overlap and balance-overlap, respectively. It can be seen from Fig. 3 (a) and Fig. 4 (a) that although the client aggregates the parameters using the attention mechanism, the server will appear when the parameters are aggregated because the data is not preprocessed before training. The case where the effect is relatively poor. This may be because the training effect of some clients is relatively poor. As a result, in the process of server aggregation, clients with poor training effects will have a negative impact on the overall training result. From Fig. 3 (b) and Fig. 4 (b), it is evident that although the training results have been improved to a certain extent, the improvement is not great. This is because the client directly performs data preprocessing before training, and the server does not assign larger weights to clients with better training results during aggregation, resulting in better-trained clients failing to drive the overall training results. Fig. 3 (c) and Fig.4 (c) are comparisons of three $FGL_{AC}$ frameworks.

### 4.3 COMPARISON BETWEEN DISTRIBUTED TRAINING AND CENTRALIZED TRAINING

To further verify the effectiveness of $FGL_{AC}$, in this subsection, three clients and one server are used for verification. Two of the clients participate in federated training, and one client uses local data sets for centralized training. Specifically, client 1 uses its training parameters for centralized training, and clients 2 and 3 perform federated training. Taking the small data set MUTAG as the test sample and Accuracy as the test index, the tests are carried out in the cases of balance-no-overlap and unbalance-no-overlap respectively, as shown in Fig. 5.

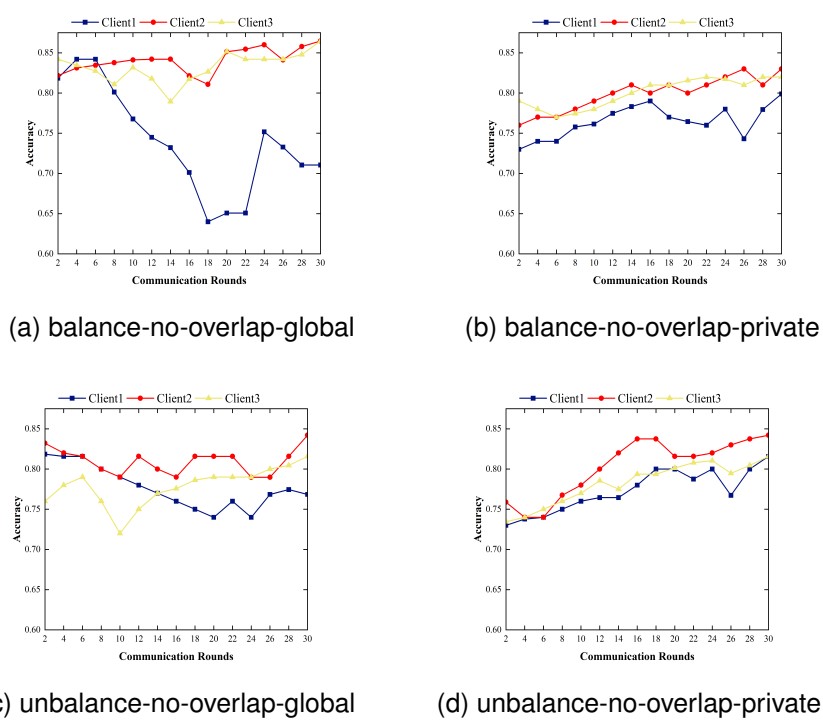

(a) balance-no-overlap-global   (b) balance-no-overlap-private

(c) unbalance-no-overlap-global  (d) unbalance-no-overlap-private

Figure 5: **Comparison between distributed training and centralized training.**

In Fig. 5, (a) and (b) are the training results of the three clients on the overall data and the rest of the private data in the case of balance-no-overlap, and (c) and (d) is in the case of unbalance-no-overlap, the training results of the three clients on the overall data and the rest of the private data. Through (a) and (c), it can be concluded that in the process of overall data training, because the client participating in the training can use the server to obtain the training parameters of the other clients, it can obtain good training in the whole training process effect. However, client 1 can only use its training parameters to train the overall data. Although it can learn some parameters through its continuous iteration, the final training result is also poor. Similarly, there will be similar training

results on other private datasets. This shows that $FGL_{AC}$ can not only bring better training accuracy than the traditional federated graph learning framework but also shows that $FGL_{AC}$ also has certain advantages for centralized model training.

## 5 CONSLUSION

In this article, a federated graph learning framework with attention mechanism and clustering algorithm is investigated, and its effectiveness is demonstrated through extensive experiments. In order to realize the framework, this paper first of all is to uses the spectral clustering algorithm to carry out a preprocessing operation on the local data held by the client before the client training, and at the same time, in the aggregation process of the server, the use of the attention method to designate different aggregation weights to various clients, to improve the training effect of the overall model. In order to better verify $FGL_{AC}$, this paper also divides the data set used for testing into four situations, making it closer to the situation in the real world. The experimental findings demonstrate that $FGL_{AC}$ will have a good improvement effect to a certain extent.

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

## A  APPENDIX

You may include other additional sections here.

### A.1  PROBLEM STATEMENT

Federated graph learning can be categorized into three groups: inter-graph federated learning (Xie et al., 2021), intra-graph federated learning (Tan et al., 2023), and graph-structured federated learning (Meng et al., 2021). This paper emphasizes on the classification of graphs, so the primary research is federated graph learning.

Inter-graph federated learning is a common learning method for federated graph learning in which each client sample is graph-structured data and the global model performs graph-level tasks. Each client holds a confidential dataset $D_k$, which contains multiple graphs $G_i$ and corresponding labels $y_i$. Due to industry competition and privacy issues in the industrial Internet, data sharing cannot be directly performed, but it can be realized under the framework of inter-graph federated learning. In this case, $D_k = (G_i^{(k)}, y_i^{(k)})$, the global model of the graph neural network can be expressed as:

$$\widehat{y}_i^{(k)} = H(X_i^{(k)}, A_i^{(k)}, W) \tag{10}$$

Where, $X_i^{(k)}$ and $A_i^{(k)}$ represent the features and adjacency matrix of the $i^{th}$ graph in the data set of the $j^{th}$ client, respectively, and $\widehat{y}$ represents the output.

The aggregation of the central server takes FedAvg as an example, and the objective function can be expressed as:

$$\min w \frac{N_k}{N} \sum_{k=1}^{K} f_k(W) \tag{11}$$

$$f_k(W) = \frac{1}{N_k} \sum_{i=1}^{N_k} L(H(X_i^{(k)}, A_i^{(k)}, W)) \tag{12}$$

Where, $f_k(W)$ is the local objective function, $L$ is the global loss function, and $N_k$ represents the number of all nodes in the data set of the $k^{th}$ client.

### A.2  CLIENT MODEL

This paper mainly focuses on the task of graph classification, that is, each client has a set of graph structure data. First, the spectral clustering explained in Section 3.2 is used to preprocess the data, and the parameters are sent to the server for local model training aggregation. The server returns the aggregated parameters to the client participating in the training through the attention mechanism in Section 3.3, and the client performs update training according to the received parameters to achieve a better global training effect. Therefore, two common GNN models are used in the model training phase of the client: GCN and Graph SAGE.

Specifically, GCN follows the strategy of neighborhood aggregation, that is, iteratively updates the representation of a node by aggregating its neighborhood representation and the information of the node itself. The $L + 1^{th}$ layer aggregation rules of GCN can be expressed as:

$$H^{(l+1)} = \sigma(\tilde{D}^{-\frac{1}{2}} \tilde{A} \tilde{D}^{-\frac{1}{2}} H^{(l)} W^{(l)}) \tag{13}$$

Where, $H^{(l)}$ is the representation matrix of nodes in layer $l$, $W^{(l)}$ is the trainable parameter matrix, $\tilde{A} = A + I_N$ is the adjacency matrix containing self-connections, and $\sigma$ is the nonlinear activation function.

Graph SAGE randomly samples the neighbors of each node in the graph and obtains the node representation by aggregating neighbor information. In this paper, the Graph SAGE pooling aggregation strategy is adopted. The $L^{th}$ layer aggregation rules of Graph SAGE can be expressed as:

$$AGGREGATE_k^{pool} = \max(\{\sigma(W_{pool}h_{u_i}^k + b),$$
$$\forall u_i \in N(v)\})$$
(14)

Where, the feature of node $v$ is expressed as $h_v^k$, $AGGREGATE_k^{pool} = \max(\{\sigma(W_{pool}h_{u_i}^k + b), \forall u_i \in N(v)\})$ is the feature of neighbor node $u_i$, and $N(v)$ is the set of neighbor nodes of node $v$. $W_{\text{pool}}$ and $b$ are the parameters of the model, used for linear transformation and bias, and $\sigma$ is the activation function. For the results of all neighbor nodes $u_i$, take the maximum value on the feature dimension.

In section 3.2, this paper uses the spectral clustering algorithm to preprocess the data set and also obtains some parameters, so these parameters can be directly used in the local model training. For example, for GCN, the spectral clustering algorithm has obtained the similarity matrix $W$ and degree matrix $D$ for the data set, and these two parameters can be directly used when the GCN model is aggregated. But for the Graph SAGE model, there is one thing that needs to be changed. In the original formula, Graph SAGE samples and aggregates the neighbor nodes of the target node. However, after preprocessing by the spectral clustering algorithm, the information of other nodes belonging to the same cluster as the target node can be obtained, which will have a better impact than the initial neighbor nodes. The changed objective function can be expressed as:

$$AGGREGATE_k^{pool} = \max(\{\sigma(W_{pool}h_k^{u_i} + b),$$
$$\forall u_i \in N_{cluster}(v)\})$$
(15)

Where, $N_{cluster}(v)$ means that the spectral clustering algorithm divides the node set in the cluster to which node $v$ belongs, rather than the set of adjacent nodes of node $v$.

## A.3 EXPERIMENT SETUP

### A.3.1 DATASETS

This work verifies the proposed $FGL_{AC}$ framework on three open datasets, namely the chemical compound domain, and biological protein domain. The first is to process the data set and use different division mechanisms to divide the data set to different clients for training. Table 3 summarizes the relevant statistics of the dataset.

Table 3: **Synopsis of datasets.**

| Field | Datasets | Graph number | Graph category | Average number of nodes | Number of node labels |
|---|---|---|---|---|---|
| Chemical Compound | MUTAG | 188 | 2 | 17.7 | 7 |
| Biological Protein | ENZYMES | 600 | 6 | 32.6 | 2 |
| | PROTEINS | 1,113 | 2 | 39.1 | 3 |

In a chemical compound dataset, each graph usually represents a compound, the nodes in the graph represent atoms, and the edges represent the real chemical bonds between atoms.

- The MUTAG data set is comprised of 188 chemical compound structure diagrams with labels categorized as mutagenic or non-mutagenic. The graph's nodes represent atoms, whereas the node labels represent the categories of atoms.

In the biological protein data set, each graph usually represents the high-level structure of a protein. The nodes in the graph represent an amino acid molecule, which represents the structural proximity between amino acid molecules. When the distance between amino acids is less than a certain threshold, the distance between nodes There are edges in between.

- ENZYMES is a data set of protein tertiary structures containing 600 enzymes. Each graph represents a protein structure, and the labels on the graphs correspond to the six enzymes' hierarchical categories.

- The 1113 graphs in the PROTEINS dataset represent proteins, and their labels are divided into two categories, representing enzymes and non-enzymes, respectively. Nodes are the secondary structure of proteins.

To better verify the effectiveness of the $FGL_{AC}$ framework, this paper processes the dataset into four test data, namely: balance-no-overlap, unbalance-no-overlap, balance-overlap, and unbalance-overlap. $FGL_{AC}$ is compared with existing methods under four different data storage situations.

### A.3.2 BASELINE

In order to more accurately assess the efficacy of $FGL_{AC}$, it is compared with four baselines for federated graph learning, including GCN-based FedAvg, SAGE-based FedAvg, GCN-based FedProx, and SAGE-based FedProx. The final results are compared with the four indicators of Accuracy, F1, Precision, and Recall.

- GCN(Kipf & Welling, 2016): When training the local model, consider the attributes of the node itself and the attributes of the adjacent nodes of the node to obtain the feature vector of the node.

- SAGE(Hamilton et al., 2017): This model contains sampling and aggregation. First, the connection information between nodes is utilized to sample neighbors, and then the information of adjacent nodes is continuously aggregated using multi-layer aggregation functions.

- FedAvg(McMahan et al., 2017): The central server of the framework aggregates local model parameters into global model parameters through weighted average aggregation.

- FedProx(Li et al., 2020): This framework introduces a regularization term to counteract the bias between global and local models by introducing a difference term between the local objective function and the global model during local training. By adjusting the hyperparameters of the regularization term, FedProx accomplishes a balance between the accuracy of the global model and the accuracy of the local model.

### A.3.3 IMPLEMENTATION

All experiments are performed on a GPU server equipped with two NVIDIA GeForce RTX 3090 GPUs and 12th Gen Intel(R) Core (TM) i9-12900K 24-core processors. The versions of Python and PyTorch are 3.8.0 and 1.11.0, respectively.

The size of the hidden layer of all models is 32, and the split of the data set is 82, 80% of which is used for model training and 20% for model testing; batch size is set to 32, and the round of federated training number is 30 rounds, and the epochs of each round are set to 70; the SGD optimizer with weight decay of 1e-4 is used, and the learning rate is 0.01. The data privacy protection mechanism is differential privacy.

### A.4 PERFORMANCE COMPARISON

In this section, the data set MUTAG is taken as an example, and the GCN-based FedAvg and SAGE-based FedProx algorithms are used as comparison objects to compare the performance of $FGL_{AC}$. The specific results are shown in Fig. 6. In this comparison experiment, the MUTAG data set is divided into four situations according to Section 4.1.1, and the indicators include four types, namely Accuracy, F1, Precision, and Recall. From the figure, it can be concluded that $FGL_{AC}$ is in the leading edge in most of the metrics in the four cases, which indicates that the method proposed in this paper is well optimized for the graph classification task of federated learning and improves the overall classification accuracy. This is because before the client participates in training, it first uses the spectral clustering algorithm to perform certain preprocessing operations on its local data set

so that it can reduce the pressure on server aggregation when participating in federated learning. At the same time, when the server aggregates, by using the attention mechanism to operate, it can maximize the training parameters of the client with a better training effect to drive the client with a poorer training effect, thereby improving the overall training results.

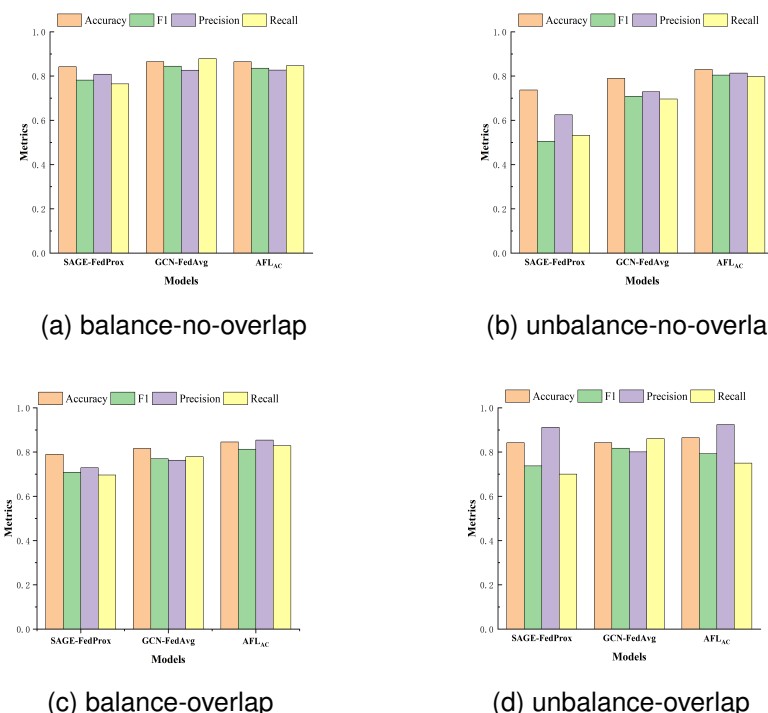

(a) balance-no-overlap

(b) unbalance-no-overlap

(c) balance-overlap

(d) unbalance-overlap

Figure 6: **Comparison of $FGL_{AC}$ and traditional federated graph learning.**

A.5 VISUALIZATION

This section takes the MUTAG dataset as an example to perform low-dimensional vector visualization tasks. It also compares $FGL_{AC}$ with traditional GCN-based federated graph learning to further verify the performance of $FGL_{AC}$. Briefly, using the node embedding vectors on the hidden layer of the last iteration of each server aggregation, the high dimensional node embeddings are transformed into low dimensional representations using the $t - SNE$ algorithm, while the $plt.show()$ method is used to display the results visually. The specific results are shown in Fig. 7.

This visualization task also divides the dataset into four cases for comparison, where Fig. 7(a) and (b) are in the balance-no-overlap case, (c) and (d) are in the unbalance-no-overlap case, and Fig. 7(e) and (f) are in the balance-overlap case, (g) and (h) are in the unbalance-overlap case. As can be seen in the figure, the performance of traditional federated graph learning is not always satisfactory in the four cases. This is because traditional federated graph learning simply aggregates the parameters uploaded by the client without considering the influence of the client with better training results on the overall training effect, and also the client is trained using the most primitive dataset without further processing of the local dataset. $FGL_{AC}$ performs better in visualizing the results of the task, and the classification results are more accurate. At the same time, the nodes of different categories are closer to each other, although some points are in the set of other points, but there is no overlapping phenomenon that is very obvious.

A.6 ATTENTION PARAMETER ANALYSIS

In order to further verify the influence of attention parameters on the overall model training during the $FGL_{AC}$ server aggregation process, this section also simulates the learning process of three

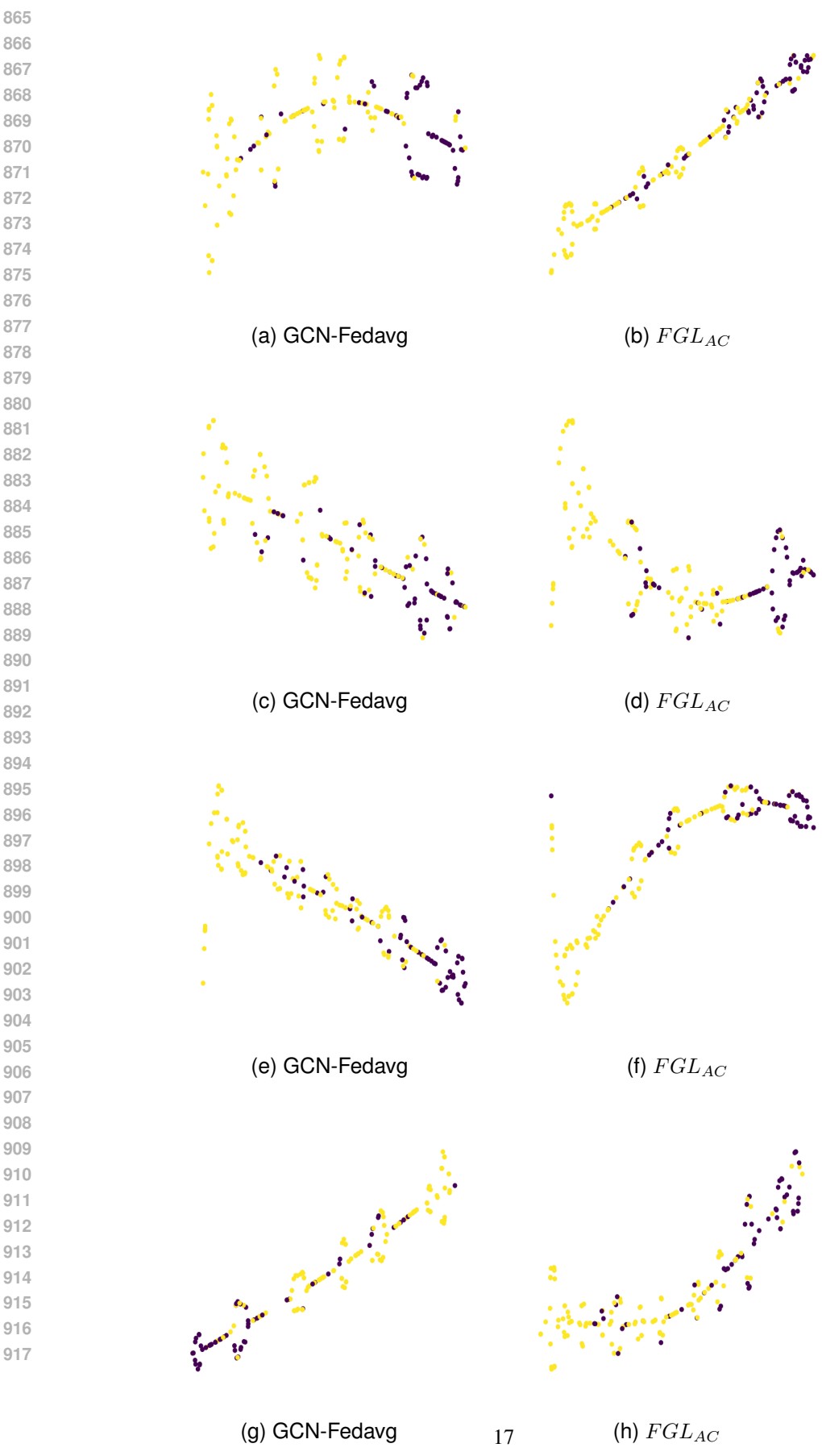

(a) GCN-Fedavg

(b) $FGL_{AC}$

(c) GCN-Fedavg

(d) $FGL_{AC}$

(e) GCN-Fedavg

(f) $FGL_{AC}$

(g) GCN-Fedavg

(h) $FGL_{AC}$

Figure 7: **Visualization analysis of $FGL_{AC}$ and GCN-Fedavg under different conditions.**

clients and one server and analyzes the changes of different client parameters in four cases. The specific result is shown in the Fig. 8.

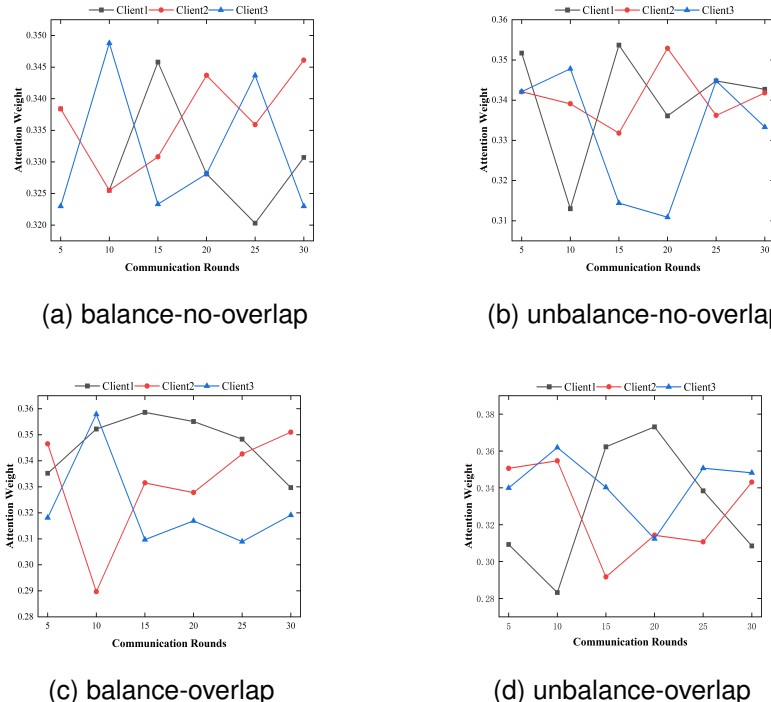

(a) balance-no-overlap           (b) unbalance-no-overlap

(c) balance-overlap           (d) unbalance-overlap

Figure 8: **Analysis of attention aggregation parameters of each client under different conditions.**

It is evident from the figure that the server uses different weights for different clients during aggregation, instead of performing average weighted aggregation like traditional federated learning. Throughout the procedure of aggregation, the server will assign different weights to different clients for their training effects according to the attention mechanism, which has the advantage of enabling the clients with better training results to occupy larger weights, bringing a positive impact on the overall training of the model. At the same time, for clients with poor training effects, the server will assign smaller weights to these clients during the aggregation process, reducing the negative impact on the overall model training. Through the proof of the above section, this mechanism is effective and can indeed improve the training effect of the model. Even with the same training effect on all clients, $FGL_{AC}$ degenerates into FedAvg.

