# OpenReview forum: "A Federated Graph Learning Framework With Attention Mechanism and Clustering Algorithm"
_ICLR.cc/2025/Conference — Submitted to ICLR 2025_

### Official Review · Reviewer_QGrg · 2024-10-23

**Soundness:** 2
**Presentation:** 1
**Contribution:** 1
**Rating:** 3
**Confidence:** 5

**Summary:**

The authors propose an attention-based aggregation scheme in a federated graph learning system and consider a data pre-processing step before training, which can increase transmission efficiency. Experimental results compared with FedAvg and FedProx are provided to show some advantages of the proposed scheme.

**Strengths:**

1. Applying attention to allocation weights for different clients seems to be promising.

2. Federated Graph Learning is a good topic.

**Weaknesses:**

1. Not a good presentation. Many long sentences or unclear expressions exist throughout the manuscript. For example, on page 9, "t can be seen from Fig. 3 (a) and Fig.4 (a) that although the client aggregates the parameters using the attention mechanism, the server will appear when the parameters are aggregated because the data is not preprocessed before training. The case where the effect is relatively poor." The second sentence has grammar errors.

2. It seems to be standard to apply attention to estimate the aggregation weight in the proposed scheme. The reviewer has found limited technical contributions or unique challenges that are solved by the proposed method.

3. Comparisons with other advanced aggregation schemes are lacking. Only Fedavg and Prox are used, where an amount of robust aggregation is available to reasonably allocate weights.

**Questions:**

1. It is not clear how the data pre-processing is done in the experimental results. Although some analyses are conducted, the advantages mentioned in the paper, transmission gain, are not provided or even discussed, which makes this contribution less important. The authors should enhance the necessity of pre-processing in graphs, which from the point of the reviewer, this step has limited technical contributions.

2. Comparisons with other advanced algorithms that allocate different weights to clients are not considered. In addition, some reported results show the proposed algorithm is even worse than Prox, which shows fewer contributions. For example, “Shapleyfl: Robust federated learning based on Shapley value,” which uses Sharply value to calculate the weights, should be compared.

---

### Official Review · Reviewer_MJ5B · 2024-10-29

**Soundness:** 1
**Presentation:** 1
**Contribution:** 1
**Rating:** 1
**Confidence:** 4

**Summary:**

This paper presents a novel method, $FGL_{AC}$, designed to enhance federated graph learning by accelerating convergence and minimizing communication overhead. The approach integrates clustering during clients' local training and employs an attention mechanism for server-side aggregation. While the experimental results demonstrate that $FGL_{AC}$ achieves superior accuracy and F1 scores compared to baseline models, the work appears to build incrementally on existing methods.

**Strengths:**

The paper effectively merges clustering algorithms with attention mechanisms, resulting in the $FGL_{AC}$ framework. Empirical results indicate that this combination outperforms baseline models, such as GCN-FedAvg, in terms of both accuracy and F1 score.

**Weaknesses:**

1. Clustering methods have been extensively explored in Graph Neural Networks (GNNs) [1]. Additionally, weight optimization for aggregation in federated learning (FL) has been addressed in prior works [2][3], which focus on aggregating model parameters. The proposed method primarily combines these two existing techniques, which may position the paper as incremental rather than offering substantial innovation.

2. The paper lacks detailed explanations of key components. For instance, the derivation of $C_{i,j}$ from $L$ is unclear, suggesting that significant steps may have been omitted. Furthermore, the role and application of the SSE metric are not adequately explained. The attention mechanism implementation also requires more clarity. Specifically, how the weights for aggregation are derived from the computed attention scores, whether there are any constraints on these weights, and the formulation of the loss function used to train the attention model.

3. The experimental settings are not sufficiently described within the main text, being relegated to the appendix. This makes it difficult to fully assess the validity of the experiments. Additionally, the term "balance-no-overlap" in Table 2 is not clearly defined. The ablation study provided is also limited, which restricts the understanding of the contribution of each component of $FGL_{AC}$.

4. The manuscript could benefit from improved clarity and organization. The discussion of related work appears somewhat disconnected, particularly regarding the implementation of the attention mechanism in federated aggregation as opposed to node aggregation in GNNs. Moreover, the paper does not sufficiently address previous approaches to weighted aggregation in FL, which could provide valuable context and strengthen the discussion.

**Questions:**

See weaknesses.

---

### Official Review · Reviewer_dZCe · 2024-11-03

**Soundness:** 2
**Presentation:** 1
**Contribution:** 1
**Rating:** 3
**Confidence:** 5

**Summary:**

This paper presents a novel federated graph classification framework that incorporates attention-based aggregation and a local clustering operation. Prior to initiating federated graph learning, each client clusters its local graph data using KNN algorithms and subsequently performs weighted aggregation. The aggregation weights for different clients are determined based on their uploaded parameters. Experimental results on three datasets demonstrate the effectiveness of the proposed method, showcasing its superior performance compared to several baseline approaches.

**Strengths:**

- The paper effectively addresses the challenges associated with graph-level federated learning for graph classification, an important and practical task. However, further exploration into more diverse or real-world datasets could enhance its applicability.

- While the model demonstrates solid performance across three commonly used datasets, expanding the evaluation to include more complex or varied datasets would provide a better understanding of its robustness and generalizability.

- Although the majority of the paper is easy to follow, certain sections could benefit from clearer explanations or more detailed descriptions to aid reader comprehension.

**Weaknesses:**

1. The motivation is confusing. (1) It is unclear why the Industrial Internet of Things (IIoT) would generate large amounts of graph data, as the paper does not provide examples or detailed explanations. (2) The rationale for pre-processing data before federated learning (FL) lacks persuasiveness and supporting evidence. For example, the paper does not demonstrate whether unprocessed graph data would significantly impact training time or performance.

2. The contribution appears limited, as the use of clustering and attention-based aggregation are common techniques in FL and are considered basic operations. The paper offers minimal novel insights or advancements to the research community.

3. The experiments are insufficient for validating the proposed model. (1) The chosen baselines are relatively weak, including GraphSAGE, GCN, FedAvg, and FedProx. Relevant federated graph classification baselines, such as those found in [1] and [2], are notably missing. (2) The experimental settings are not clearly described, creating ambiguity around configurations such as balance-no-overlap and data splitting strategies.

4. The paper contains multiple formatting and grammatical errors, including issues with capitalization and improper formatting of figures and tables. A thorough review by a professional editor is recommended to improve readability.

- References:

[1] Xie H, Ma J, Xiong L, et al. Federated graph classification over non-iid graphs. Advances in Neural Information Processing Systems, 2021, 34: 18839-18852.

[2] Tan Y, Liu Y, Long G, et al. Federated learning on non-iid graphs via structural knowledge sharing. Proceedings of the AAAI Conference on Artificial Intelligence, 2023, 37(8): 9953-9961.

**Questions:**

- Could you elaborate on why the Industrial Internet of Things (IIoT) would generate a large amount of graph data? The paper does not provide examples to illustrate this point.

- What is the rationale behind the need to process graph data before applying federated learning (FL)? The reasoning provided is not entirely convincing and seems to lack supporting evidence.

---

### Official Review · Reviewer_CmJ5 · 2024-11-05

**Soundness:** 2
**Presentation:** 2
**Contribution:** 1
**Rating:** 3
**Confidence:** 4

**Summary:**

The paper proposes a Federated Graph Learning framework called FGLAC, designed for secure and efficient graph data processing in the context of the Industrial Internet of Things (IIoT).

**Strengths:**

The authors employ a spectral clustering algorithm to preprocess the data at the client level before federated training. This preprocessing reduces data dimensionality, easing the computational load and potentially enhancing classification accuracy. During aggregation, the server uses an attention mechanism to assign dynamic weights to the parameters from each client. This allows the global model to favor parameters from clients with higher-quality training, thus improving overall accuracy.

**Weaknesses:**

There are multiple aspects that the paper need to strengthen:
1. The baselines used in this paper are old and there is a lack of most up-to-date baselines. The novelty of the paper needs to be justified by comparing with these most recent work on federated GCN.

2. Although spectral clustering is used for preprocessing, the choice of spectral clustering over other clustering methods (e.g., k-means, hierarchical clustering) could be more rigorously justified. Including a comparison with alternative clustering techniques could strengthen the evidence of its effectiveness.

3. The paper seems to be an empirical integration of clustering and attention mechanisms for federated GCN with little theoretical backgrounds to demonstrate why this would work.

**Questions:**

1. Add recent baselines for comparison to empirically demonstrate the benefits of the proposed algorithm.

2. Justify the novelty of the paper with most recent work on federated GCN. The paper should compare with some closely related papers:
a. FedGCN: Convergence-Communication Tradeoffs in Federated Training of Graph Convolutional Networks. NeuIPS 2023
b. FedGCN: A Federated Graph Convolutional Network for Privacy-Preserving Traffic Prediction. IEEE TSC.
c. A federated graph neural network framework for privacy-preserving personalization. Nature communications.
d. FedGCN: Convergence-communication tradeoffs in federated training of graph convolutional networks. NeuIPS 2024.
e. One Node Per User: Node-Level Federated Learning for Graph Neural Networks. NeuIPS workshop 2023.

**Details Of Ethics Concerns:**

Not applicable.

---

### Meta-Review · Area_Chair_Amwe · 2024-12-15

**Metareview:**

The paper presents FGLAC, a federated graph learning framework using attention mechanisms and clustering for IIoT data, claiming improved model training effects.

Strengths: Innovative combination of clustering and attention in federated learning; shows performance improvements over baselines.

Weaknesses: Outdated baselines, lack of theoretical background, poor presentation, and limited novelty.

Decision: Reject. The paper needs updates with recent baselines, clearer motivation, better theoretical support, and improved presentation quality.

**Additional Comments On Reviewer Discussion:**

There was no rebuttal

---

### Decision · Program_Chairs · 2025-01-22

Reject